# Experimental Characterization of the Drying of Kampot Red Pepper (*Piper nigrum* L.)

**DOI:** 10.3390/foods9111532

**Published:** 2020-10-24

**Authors:** Elen Morm, Khamphon Ma, Sovivort Horn, Frédéric Debaste, Benoit Haut, Sokneang In

**Affiliations:** 1Faculty of Chemical and Food Engineering, Institute of Technology of Cambodia, Blvd. of the Russian Federation, Phnom Penh P.O. Box 86, Cambodia; morm.elen@itc.edu.kh (E.M.); Ma.Khamphon@itc.edu.kh (K.M.); Horn.sovivort@itc.edu.kh (S.H.); in@itc.edu.kh (S.I.); 2Transfers, Interfaces and Processes, Université libre de Bruxelles, Av. F.D. Roosevelt 50, ULB CP165/67, 1050 Brussels, Belgium; fdebaste@ulb.ac.be

**Keywords:** boiling pretreatment, kampot pepper, *Piper nigrum* L., drying, total phenolic content, piperine

## Abstract

The objective of this work is to provide new insights into the mechanisms taking place during the drying of the mature grains of Kampot pepper, a cultivar of pepper (*Piper nigrum* L.), which is produced in the Kampot Province, Cambodia. Indeed, even if the Kampot pepper is recognized for its organoleptic qualities, no research works were dedicated to the drying of its mature grains, in order to yield red pepper. Experiments with different pretreatment and drying conditions were performed. The results of these experiments were analyzed, regarding the drying kinetics, the color of the dry product, and the degradation of the bioactive compounds during the drying. Regarding these bioactive compounds, several parameters were considered: the total phenolic content, the total flavonoid content, and the piperine content. The results show that the Kampot mature pepper is prone to alterations when dried at a temperature of $55{\;}^{{}^{\circ}}$C or $65{\;}^{{}^{\circ}}$C: the color, the total phenolic content, and the flavonoid content are significantly altered, while the piperine content, important for the pungency of this spice, seems unaltered. Raising the temperature leads to more important degradations. However, performing a pretreatment by dipping the pepper grains into boiling water appears to significantly reduce these alterations and, concomitantly, to accelerate the drying. As a conclusion of the analysis of the results, it can be stated that, to increase the product quality, it is recommended to pretreat the pepper by dipping it into boiling water during 5 min., before drying at $55{\;}^{{}^{\circ}}$C.

## 1. Introduction

Pepper is not only used for enhancing the food flavor, but also for food preservation and for pharmacological purposes due to its content in various bioactive and nutritional compounds. Those compounds include phenolic compounds (for instance, flavonoid compounds), vitamin C, essential oils, minerals, carbohydrates, proteins, fats, fibers, and piperine. The latter, giving its characteristic pungency to the pepper, is an alkaloid molecule that is known for presenting anti-inflammatory, anti-diarrheal and anti-hypertensive properties [1,2].

The Kampot pepper is a cultivar of pepper (*Piper nigrum* L., from the Piperaceae family), which is produced in the Kampot Province, Cambodia. The export of Kampot pepper to Europe has been first reported in the years 1870, during the French Protectorate, and it has grown in importance since then, except during the Khmer Rouge rule. Because of the reputation of its organoleptic quality, it was recently registered as a Protected Geographical Indication [3].

In Cambodia, four types of Kampot pepper are found on the market: green, black, red, and white pepper. The difference between these products comes from the harvest period and the transformation operations. Green pepper is the fresh grains, harvested before maturity but when the fruit kernel is well formed. It is served in fresh (i.e., undried) form or preserved in saline or oil. The three other types of pepper are dried. Black pepper is the result of the drying of the grains harvested before maturity (yellow to green color). Red pepper is the result of the drying of mature grains, which present a red color before and after the drying (see Figure 1). Finally, white pepper is the product that is obtained by removing the exocarp of mature grains prior to their drying. The grains have a spherical shape with a diameter around 5–7 mm. They shrink during their drying, to reach a diameter around 4–5 mm. The mass of a single grain is about 0.1 g.

Most fresh vegetables and fruit have a short shelf-life, because they present a relatively high moisture content, promoting microbial spoilage and the development of detrimental enzymatic reactions. Consequently, these food products cannot be stored for a long period, and drying is one of the traditional techniques used to improve their shelf-life.

Traditionally, the drying of Kampot pepper is performed during three to four days, depending on the weather conditions, by spreading the grains on the ground, outdoor, exposed to direct sunlight. Usually, before the drying, the grains are soaked in boiling water for a few minutes. In this paper, this operation is referred to as the pretreatment. It is thought to play an important role in reducing the drying time, ensuring the microbiological quality and inactivating the the polyphenol oxidases (PPOs), enzymes causing the browning of the pepper during the drying. Achieving the desired color of the product plays indeed an important role towards its acceptability by the consumers [4].

Even if the Kampot pepper is recognized for its organoleptic qualities, no research works were dedicated to the drying of mature grains, in order to yield red pepper. The drying of *Piper nigrum* L. grains has been studied, but only for the production of black pepper [5]. Regarding the drying of mature grains to yield red pepper, all of the research that has been conducted was dedicated to *Capsicum annuum* L., from the Solanaceae family [6,7,8,9], which is a species that differs strongly from *Piper nigrum* L.

Therefore, the objective of this work is to provide new insights into the mechanisms taking place during the drying of Kampot mature pepper grains. More precisely, our objective is to analyze the influence of the pretreatment and of the drying temperature on the drying kinetics (i.e., the rate of water removal) and on the color and nutritional quality of the dry pepper.

For this purpose, experiments with different pretreatment durations and drying temperatures were performed. The results of these experiments were analyzed, regarding the drying kinetics, the color of the dry product, and the degradation of the bioactive compounds during the drying. Regarding the bioactive compounds, several parameters were investigated: the piperine content, the total phenolic content, and the total flavonoid content.

## 2. Material And Methods

### 2.1. Plant Material

Fresh Kampot mature pepper grains (red color) were obtained, 24 h before each experiment, from La plantation pepper farm, Kampot province, Cambodia. The samples were stored in an icebox while being transported to the laboratory and, once there, immediately placed at $4{\;}^{{}^{\circ}}$C until experimentation.

### 2.2. Chemicals

The HPLC grade solvents, including methanol (Merck, Kenilworth, NJ, USA), acetonitrile (Merck), HPLC water grade (Merck) and acetic acid glacial (Fisher, Hampton, NH, USA), and the chemical reagents, including potassium chloride (Merck), sodium acetate anhydrous (Scharlau, Barcelona, Spain) and Folin-Ciocalteu reactant (Sigma–Aldrich, St. Louis, MO, USA), were purchased from Dynamic Scientific Co. Ltd. (Phnom Penh, Cambodia). A piperine standard (>99%, Sigma-Aldrich) was purchased from Sigma–Aldrich (Brussels, Belgium).

### 2.3. Boiling And Drying

For each drying experiment, fresh grains were first washed with distilled water thrice in order to remove the impurities. Then, they were soaked in boiling water for 5, 10 or 15 min. These durations correspond to usual practices in the fields in Cambodia for red pepper production. A large excess of water was used (about 30 times the mass of the grains). It was checked that such a large excess allowed for the temperature of the water (i.e., 100${\;}^{{}^{\circ}}$C) to remain constant when the grains were introduced in the boiling water. Drying experiments were also performed with unboiled grains (i.e., without pretreatment).

Subsequently, for each drying experiment, a sample of the grains (approximately 30 g) was dried in an oven (Memmert, Germany), at $55{\;}^{{}^{\circ}}$C or $65{\;}^{{}^{\circ}}$C. The characteristics of this oven are given in Appendix A. During the first 8 h of the drying, the sample was weighted every hour. Afterwards, the sample was left overnight in the same oven, at the same temperature and with the same ventilation. Subsequently, on the next morning, we systematically checked that the drying had stopped, by checking that the sample had reached a constant mass, which was measured. At the end of the drying, the sample still contains some residual water, as it is at the thermodynamic equilibrium with the drying air. Therefore, the sample was subsequently freeze-dried before further analysis. Each experiment (i.e., at given temperature and boiling pretreatment duration) was done in duplicate.

The mass *m* of the sample measured at any time during a drying experiment can be transformed into its moisture content *X* (in kg of water per kg of dry matter), if its initial moisture content $X_{0}$ (i.e., its moisture content at the beginning of the drying, after a possible pretreatment) and its initial mass $m_{0}$ are known:(1)$$X=\frac{m(1+X_{0})}{m_{0}}-1$$

In order to be able to transform the mass of the sample into its moisture content, its initial moisture content $X_{0}$ was determined in parallel with each drying experiment by putting other grains from the same pretreatment batch in another oven, at 105${\;}^{{}^{\circ}}$C for 24 h. It was checked that this duration is enough to yield a constant mass of the grains. The initial mass of these other grains $m_{0}^{*}$ and their final mass $m_{f}^{*}$ (i.e., their mass after spending 24 h in the oven, which is assumed to be the mass of dry matter in these grains due to the high temperature used) were measured with a precision balance (ES 360 series, 125 SM model, Precisa gravimetric AG, Dietikon, Switzerland). $X_{0}$ was then evaluated as:(2)$$X_{0}=\frac{m_{0}^{*}-m_{f}^{*}}{m_{f}^{*}}$$

### 2.4. Analytical Methods

#### 2.4.1. Color Measurement

Color measurements were conducted on the fresh grains (i.e., without pretreatment and drying) and, for each drying experiment, on the dry grains (i.e., at the end of the drying experiment), using a Chroma meter (Konica minolta, CR-400) that was equipped with a pulsed xenon lamp. The observer was the 2${}^{{}^{\circ}}$ 109 matches CIE 1931 Standard. The observer was calibrated before measurements using the corresponding standard white plate calibrator CR-A430. The results were expressed in $L^{*}$, $a^{*}$ and $b^{*}$ values, the color coordinates in the CIELAB color space [10]. The grains were not grounded before measuring their color. Indeed, our objective was to characterize the color of the whole dry grains, which is an important parameter for the consumer when it is sold as is. For measurement, the grains were placed in a granular attachment pad CR-A50. In order to compare the color of dry and fresh grains, the total color difference ΔE was determined for each drying test:(3)$$\Delta E=\sqrt{{\left(L^{*}-L_{0}^{*}\right)}^{2}+{\left(a^{*}-a_{0}^{*}\right)}^{2}+{\left(b^{*}-b_{0}^{*}\right)}^{2}}$$
where $L^{*}$, $a^{*}$, $b^{*}$, and $L_{0}^{*}$, $a_{0}^{*}$, $b_{0}^{*}$ are the Chroma values of the dry and fresh grains, respectively.

For each drying experiment, $L^{*}$, $a^{*}$ and $b^{*}$ were measured three times. Hence, we obtained six values of these parameters (and of ΔE) for given drying temperature and boiling time, as the drying experiments were done in duplicate. For the fresh grains $L_{0}^{*}$, $a_{0}^{*}$ and $b_{0}^{*}$ were measured 6 times. The data were analyzed with an analysis of variance (ANOVA) test.

#### 2.4.2. Extraction

For further analysis, a sample of fresh grains (i.e., without pretreatment and drying) and each freeze-dried sample were grinded with a blender (International, Cambodia) until a powder was obtained. Before reduction to powder, the freeze-dried samples were defrosted for three hours at room temperature. Subsequently, for each sample, about 0.5 g of the powder was put in a test tube with 10 mL of methanol. Subsequently, the suspension was placed in an ultrasonic tank for 20 min. After that, the suspension was centrifuged at 2000 rpm for 10 min. (Hettich 1206, ROTOFIX 32A centrifuge, Tuttlingen, Germany). The supernatant was set aside and the solid residue was again placed in a test tube with 10 mL of methanol. These operations (ultrasonic tank, centrifugation, set aside of the supernatant, addition of 10 mL of methanol to the solid residue) were repeated three times. The supernatants collected after the three centrifugations were then put in a rotary evaporator (IKA RV10 Rotary Evaporator with HB10 Bath, Staufen, Germany) until dryness. The obtained product is further referred to as the extract.

#### 2.4.3. Piperine Content Measurement

The piperine content of fresh or dry grains was determined by using High Performance Liquid Chromatography (HPLC, Shimadzu, LC 2010A, Kyoto, Japan), with an auto sampler and a UV-detector, on the corresponding extract. This analysis was performed following a method developed by Upadhyay et al. [11], with some modifications. The stationary phase of the HPLC was a C18 column (5 μm particle size, 250 × 4 mm), while the mobile phase was an isocratic mixture of acetonitrile, water and acetic acid (60:39.5:0.5). The UV measurement was performed at 340 nm. The velocity of the mobile phase was 1 mL/min. The run time was 10 min. For each run, 10 μL of solution was injected in the HPLC column. The retention time of piperine was about 6 min. Appendix B presents typical chromatograms.

In order to obtain a calibration curve, a standard stock solution was prepared by dissolving 10 mg of a piperine standard in 10 mL of methanol. From this stock solution, a calibration curve was obtained from measurements at concentrations of 15 ppm, 25 ppm, 50 ppm, 70 ppm, and 90 ppm. Each injection was duplicated using the auto sampler. The full procedure was performed five times in series, based on the same stock solution, yielding a regression coefficient $R^{2}$ above 0.95.

For the quantification of the piperine in fresh or dry grains, a known mass of the corresponding extract was dissolved in 10 mL of methanol and then diluted with methanol to obtain a final concentration around 0.1 mg of extract/mL. The diluted sample was filtered with a 0.45 μm filter before injection in the HPLC column. The measured concentration was then converted into the piperine content of the grains (in mg of piperine per g of dry mass of the grains), via its multiplication by an appropriate factor.

For each drying experiment, the piperine content was measured six times (i.e., the whole procedure was conducted 6 times starting from the extract). Hence, we obtained 12 values of this parameter for given drying temperature and boiling time, as the drying experiments were done in duplicate. For the fresh grains, the piperine content was measured 12 times. The data were analyzed with an ANOVA test.

#### 2.4.4. Total Phenolic Content Measurement

The total phenolic content (TPC) of fresh or dry grains was determined by the application of the Folin-Ciocalteu method on the corresponding extract. The analysis was performed following the procedure developed by Mediani et al. [12]. A known mass of the extract (around 0.07 g) was solubilized in 10 mL of methanol. 0.3 mL of this solution was then mixed in a test tube with 1.5 mL of a 10-times diluted Folin-Ciocalteu reagent, in order to reach a concentration around 0.2 mg of extract/mL. After incubation for 10 min, 1.2 mL of sodium carbonate (7.5% *w*/*v*) was added to the mixture. The mixture was then shaken with a vortex mixer in order to obtain homogeneity. The resulting solution was incubated in darkness for 30 min. before being placed in a UV/VIS spectrophotometer (BK-D580, Zhangqiu, China) for the measurement of its absorbance at a wavelength of 765 nm. The measured absorbance was then converted into the TPC of the mixture, expressed as its gallic acid equivalent (GAE) concentration, in mg of GAE/mL. For this purpose, a calibration curve, which was obtained by the application of this protocole to solutions of known concentrations of gallic acid, was used. The concentrations considered for the calibration were 5–50 mg/mL and a blank solution. Finally, the TPC of the mixture was converted into the TPC of the grains (in mg of GAE per g of dry mass of the grains), via its multiplication by an appropriate factor.

For each drying experiment, the TPC was measured six times (i.e., the whole procedure was conducted six times starting from the extract). Hence, we got 12 values of this parameter for given drying temperature and boiling time, as the drying experiments were done in duplicate. For the fresh grains, the TPC was measured 12 times. The data were analyzed with an ANOVA test.

#### 2.4.5. Total Flavonoid Content Measurement

The total flavonoid content (TFC) of fresh or dry grains was determined with the aluminum chloride colorimetric assay, following the procedure developed by Kamtekar et al. [13], applied on the corresponding extract. A known mass of the extract (around 0.07 g) was solubilized in 10 mL of methanol. 1 mL of the resulting solution was then introduced into a test tube with 4 mL of distilled water and 0.3 mL of a 5% (*w*/*v*) sodium nitrite solution. After 5 min., 0.3 mL of a 10% (*w*/*v*) aluminum chloride solution was added to the solution and, after 6 min., 2 mL of a 1 M sodium hydroxide solution was added to the mixture. The mixture was finalized by adding distilled water, in order to obtain a volume of 10 mL. Subsequently, it was placed in a UV/VIS spectrophotometer (BK-D580, Zhangqiu, China) for the measurement of its absorbance at a wavelength of 510 nm. The measured absorbance was then converted into the TFC of the mixture, expressed as its quercetin equivalent (QE) concentration, in mg of QE/mL. For this purpose, a calibration curve, which was obtained by the application of this protocole to solutions of known concentrations of quercetin, was used. The concentrations considered for the calibration were 0.2–0.9 mg/mL and a blank solution. Finally, the TFC of the mixture was converted into the TFC of the grains (in mg of QE per g of dry mass of the grains), via its multiplication by an appropriate factor.

For each drying experiment, the TFC was measured six times (i.e., the whole procedure was conducted six times starting from the extract). Hence, we obtained 12 values of this parameter for given drying temperature and boiling time, as the drying experiments are done in duplicate. For the fresh grains, the TFC was measured 12 times. The data were analyzed with an ANOVA test.

### 2.5. Drying Kinetic Constant

For each drying experiment, the kinetics of the drying was modeled using a classical Newton law [14]:(4)$$\frac{X-X_{r}}{X_{0}-X_{r}}=\exp\left(-k\;t\right)$$
where *X* is the moisture content of the sample at time *t*, $X_{0}$ its initial moisture content (i.e., at the beginning of the drying, after the pretreatment), $X_{r}$ its residual moisture content (i.e., at the end of the drying, when the sample is at thermodynamic equilibrium with the drying air), and *k* a drying kinetic constant. For each drying experiment, *k* was determined by a linear regression of the experimental values of $\ln(\frac{X-X_{r}}{X_{0}-X_{r}})$ as a function of *t*. All of the linear regressions show a $R^{2}$ value above 0.95, highlighting the acceptable fit of the data by the model.

## 3. Results And Discussion

### 3.1. Fresh Product Properties

Table 1 summarizes the values of different properties of the fresh Kampot mature pepper. For example, an average piperine content of 38.5 mg/g was measured; this value is very close to the value of 38.1 mg/g obtained by Liu et al. [15], also for fresh Kampot mature pepper. It can also be mentioned that the average value that was obtained for the TPC (9.1 mg GAE/g) is of the same order of magnitude than the value of 4 mg GAE/g obtained by Shan et al. [16], also for fresh Kampot mature pepper. The differences between these two values is probably linked to differences in the state of maturity of the pepper and/or to differences in the measurement methods. No TFC measurement expressed in mg of QE/g was found in the literature for the fresh Kampot mature pepper.

### 3.2. Drying Kinetics

The time evolution of the grain moisture content during the drying and the identified values of the drying kinetic constant *k* (average values and standard deviations) are presented, for the different drying experiments ($55{\;}^{{}^{\circ}}$C or $65{\;}^{{}^{\circ}}$C; 0, 5, 10, or 15 min. of boiling) in Figure 2 and Figure 3, respectively.

It can be observed in Figure 2 that the pretreatment has a slight influence on the initial moisture content of the grains: it is increased from 1.3 on a dry basis for the unboiled grains to approximately 1.5 on a dry basis for the grains boiled for 10 or 15 min.

As expected, the results that are presented in Figure 2 show that an increase of the drying temperature leads to a smaller value of the water content of the grains after 8 h of drying (i.e., it leads to an increase of the drying rate). These results also show that, when a pretreatment is performed, the water content of the grains after 8 h of drying is smaller than the one obtained without pretreatment. After 8 h of drying, the lowest moisture content is reached for the sample dried at $65\;{\;}^{{}^{\circ}}$C, with a pretreatment of 15 min. For this experiment, the moisture content after 8 h of drying is about 0.2 kg of water per kg of dry matter. The values of the residual moisture content $X_{r}$ show no clear dependance upon the temperature and the boiling time. An average value of 0.02 kg of water per kg of dry matter is measured, with a standard deviation of 0.01 kg of water per kg of dry matter.

Coherently, the drying temperature appears to significantly influence *k*; the corresponding ANOVA test on the *k* values gives a *p*-value < 0.01. *k* is almost doubled when the temperature is increased from $55{\;}^{{}^{\circ}}$C to $65{\;}^{{}^{\circ}}$C (see Figure 3). Similarly, in their study of the drying of Sri Lankian unripe *Piper nigrum* in a spouted bed, Jayatunga et al. [5] found a 75% increase of the drying kinetic constant with a $10{\;}^{{}^{\circ}}$C increase of the drying temperature. This dependence of *k* on the drying temperature is due to the increase, with the temperature, of both the saturation pressure of the water in the air (i.e., the driving force for the drying) and the effective diffusion coefficient of the water molecules in the grains.

The results that are presented in Figure 3 show that the pretreatment has also an influence on the drying kinetic constant. For both drying temperatures, an increase of *k* is observed when the grains are boiled before the drying, when compared to unboiled grains. The boiling probably alters the grain structure, “opening” the pores, leading to an increased mobility of the water molecules inside the grains. The statistical analysis of the results also show that, at a given drying temperature, the values of *k* for a boiling of 5, 10, and 15 min. are not significantly different (*p*-value = 0.28); after 5 min of boiling, any additional boiling time will not increase significantly the subsequent drying rate. As a conclusion, from an energetic economy point of view, it seems interesting to boil the grains before the drying, but not more than 5 min.

### 3.3. Color Degradation by the Drying

For each drying experiment, Figure 4 presents the ratios of the values of the parameters $L^{*}$, $a^{*}$, and $b^{*}$ for the grains after their drying to the values of these parameters for the fresh grains (i.e., unboiled and undried), $L_{0}^{*}$, $a_{0}^{*}$, and $b_{0}^{*}$. We see on this figure that the color of the grains is significantly altered by the drying. Indeed, $L^{*}/L_{0}^{*}$, $a^{*}/a_{0}^{*}$ and $b^{*}/b_{0}^{*}$ take values significantly smaller than 1, whatever the drying conditions (for instance, they are around 0.75 for a drying with a pretreatment). Moreover, we see that the color of the grains is better preserved at 55${\;}^{{}^{\circ}}$C than at 65${\;}^{{}^{\circ}}$C (larger values of $L^{*}/L_{0}^{*}$, $a^{*}/a_{0}^{*}$ and $b^{*}/b_{0}^{*}$ at 55${\;}^{{}^{\circ}}$C than at 65${\;}^{{}^{\circ}}$C, except for $a^{*}/a_{0}^{*}$ when the boiling time is 5 min.). Finally, it also clearly observed in Figure 4 that the boiling pretreatment allows increasing the preservation of the color, but no significant difference is observed between the results obtained for the three boiling times. Therefore, a pretreatment of 5 min. in boiling water seems to be recommended for limiting the alteration of the color during the drying.

The corresponding values of ΔE, calculated while using Equation (Equation 3), are presented in Figure 5. Please note that the y-axis of this figure is not starting from zero; it can lead to think that the dispersion is high, although it is not (only a few percent). In agreement with the results presented in Figure 4, it can be observed that, at both drying temperatures, the boiling leads to values of ΔE significantly smaller (i.e., less color change) than those obtained for grains that did not undergo boiling. However, the statistical analysis of the results show that the duration of the boiling does not have a significant impact on ΔE (*p*-value = 0.17), at least for the tested values of the boiling duration. On the other hand, for a given pretreatment time, the statistical analysis shows that a higher drying temperature leads to a significantly (*p*-value < 0.01) higher value of ΔE. This again shows that a high drying temperature is detrimental to the preservation of the color of the product.

In most food products, the color changes, when exposed to high temperature, can be related to enzymatic browning caused by PPOs and/or to chemical browning caused by Maillard reactions. Here, the boiling step probably allows for a partial or total inactivation of the PPOs in the grains, reducing the amplitude of the color change during the drying [17]. The fact that a significant color change is still present, even when a boiling is performed can be attributed to a remaining activity of the PPOs and/or to Maillard reactions. These two brownings are known to be accelerated when the temperature is raised; this might explain the difference observed between the drying at $55{\;}^{{}^{\circ}}$C and at $65{\;}^{{}^{\circ}}$C. The fact that the boiling duration has no significant impact on the color change is probably linked to the fact that the fraction of the PPOs that is inactivated during the boiling is similar for the different boiling conditions. This could possibly be attributed to an inactivation of the fraction of the PPOs that is the most heat sensitive in less than 5 min. at $100{\;}^{{}^{\circ}}$C.

No data on PPOs inactivation in *Piper nigrum* were found in the literature. Yet, PPOs exposure to temperature above $75{\;}^{{}^{\circ}}$C is known to destroy the native configuration of the enzymes, leading to a loss of reactivity [18]. Accordingly, Weil et al. [19] observed a significant influence of a boiling pretreatment on the preservation of the color of *Piper borbonense* grains after their drying. They concluded that the boiling inactivates the PPOs causing browning reactions during the drying. Thus, a similar behavior seems a plausible assumption for *Piper nigrum*.

### 3.4. Influence of the Boiling and the Drying Conditions on the Piperine Content

Table 2 and Figure A4 in Appendix C present the ratio of the piperine content of dry grains to the piperine content of fresh grains, as a function of the boiling time, for the two drying temperatures: $55{\;}^{{}^{\circ}}$C and $65{\;}^{{}^{\circ}}$C. The results and their statistical analysis show that the drying, with or without pretreatment, does not seem to influence the piperine content of the grains (*p*-value = 0.32). These results are totally coherent with those that were obtained in several studies, when considering pepper with different origins [19,20,21]. This good conservation of the piperine is probably due to the fact that the piperine molecules are mostly located in the inner core of the grains, rather than in the skin [22].

### 3.5. Influence of the Boiling and the Drying Conditions on the Tpc and Tfc

Table 3 and Table 4 and Figure A5 in Appendix C present the ratios of the TPC and TFC of dry grains to the TPC and TFC of fresh grains (i.e., unboiled and undried), for the different drying temperatures and boiling times (including no boiling).

The results regarding the TPC show large standard deviations (an average of 22% for all of the drying conditions); this deserves some comments. First, it should be kept in mind that the results presented in Table 3 and Table 4 are ratios. Therefore, the relative error on these ratios is twice the one of a TPC/TFC measurement. Zarai et al. [23] performed TPC measurements for pepper grains. They report standard deviations of their results between 2% and 10%, which is coherent with our results. Moreover, it is important to highlight that the important standard deviation of the results regarding the TPC does not come from the limited number of drying tests. It rather comes from the fact that, when, after a single drying experiment, the TPC measurement is repeated six times on the resulting extract, the results obtained show a large dispersion. It is due to the difficulty of applying the Folin–Ciocalteu method in the lab in Cambodia (no air conditioning, large variation of the room temperature and humidity within hours...). This affirmation is supported by two tables in appendix (Table A1 in Appendix D). They provide, for each drying test at $65{\;}^{{}^{\circ}}$C and for each boiling time, the average and the standard deviation of the 6 evaluations of the TPC ratio. These two tables can be compared with Table 3 in the core of the text (its right part) that gives, at $65{\;}^{{}^{\circ}}$C and for each boiling time, the average and the standard deviation of the 12 evaluations of the TPC ratio (so when the results from the two tests are analyzed together). It can be observed that these three tables are quite close to each other. It is because, when an experiment is performed, the drying conditions (boiling duration, drying temperature, ventilation rate) are very well controlled and the experiment involves approximately 300 grains (making the extract independent of the natural variability of the product). This good control of the conditions and independence on the natural variability of the product (plus the high precision of mass and time measurements) explain why, in Figure 2 and Figure 3, the dispersion of the results regarding the drying kinetics is very small. Accordingly, basically, we can expect that the extracts that were obtained at the end of the two drying experiments with the same drying conditions are really close to each other. Consequently, for given drying conditions, an additional drying experiment would not reduce the dispersion. Of course, it would give 18 values of the TPC instead of 12 and make the statistical tests more robust but, despite the dispersion on the TPC results and as presented below, the statistical analysis of the results of the two drying experiments allows drawing significant conclusions.

The results and their statistical analysis show that the drying temperature has no significant influence on the TPC and TFC (*p*-value = 0.11) of the dry grains (at least for the two temperatures considered), while the pretreatment has a significant impact on these contents (*p*-value = 0.02). Without pretreatment, the drying reduces the TPC and the TFC by a factor 2 (see Table 3, Table 4 and Figure A5). When a pretreatment is performed, the results show that the TPC and TFC of the dry grains are even higher than the TPC and TFC of the fresh grains. A good conservation of these polyphenol contents after the drying when a pretreatment is performed was expected. Indeed, it has already been mentioned that soaking the grains into boiling water for a few minutes probably leads to an inactivation of the PPOs and, thus, prevents degradations of the polyphenols during the subsequent drying [24].

The increase of the TPC and TFC after the drying may be related to the fact that the pretreatment leads to an improvement in the ability to extract the polyphenols from the grains, by “opening” their porous structure. If such an effect was, to the authors knowledge, not described for *Piper nigrum*, it is described in many other food matrices [25], including fruit based products [26].

However, the significance of these TPC and TFC increases due to the boiling is questionable, mainly for the TPC for which a large variability of the experimental results is observed. Regarding the TFC, its increase due to the drying (after pretreatment) is about 30%, whatever the pretreatment duration (among the conditions tested). These results are coherent with results that were previously published regarding hot chili pepper (*Capsium frutescens var. sina*) and sweet pepper (*Capsium annuum var. goduion*) [27].

## 4. Conclusions

The results that were obtained in this work show that the Kampot mature pepper (red color) is prone to alterations when dried at a temperature of $55{\;}^{{}^{\circ}}$C or $65{\;}^{{}^{\circ}}$C: the color, the total phenolic content and the flavonoid content are significantly altered, while the piperine content, which is important for the pungency of this spice, seems unaltered. Raising the temperature leads to more important degradations.

Realizing a pretreatment by soaking the pepper grains into boiling water appears to significantly reduce these alterations and, concomitantly, to accelerate the drying. Indeed, the boiled samples have a less degraded color after the drying, while their total phenolic content and the total flavonoid content are preserved or might even be increased. Two phenomena could explain these impacts of the boiling pretreatment on the drying:the inactivation, by the boiling, of the PPOs. Indeed, these enzymes are known to be linked to the color and polyphenol degradation in a product, when exposed to high temperature; and,the degradation of the solid pepper matrix to a more open structure, leading to enhanced mobility of the polyphenols (and, hence, to an increased ability to extract them) and of the water molecules.

These phenomena are seen already with the smallest tested boiling time of 5 min. and do not appear to have a significantly larger influence on the drying when the boiling is prolonged up to 15 min. Consequently, to increase the product quality, it is recommended to pretreat the pepper by dipping it into boiling water during 5 min., before drying at $55{\;}^{{}^{\circ}}$C. The choice of this temperature is motivated by the fact that the color of the grains is better preserved at 55${\;}^{{}^{\circ}}$C than at 65${\;}^{{}^{\circ}}$C (while the TPC, TFC, and piperine content of the dry grains are not significantly influenced by the drying temperature). However, it is worth mentioning that this study did not take into account the microbiological aspects that might require, for food safety reasons, longer boiling.

## Figures and Tables

**Figure 1 foods-09-01532-f001:**
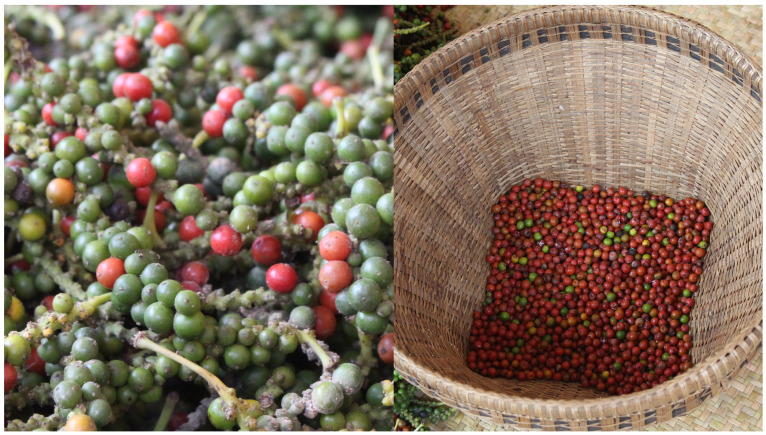
Left: pepper grains showing different level of maturity. Right: harvested mature pepper grains. Pictures taken by the authors.

**Figure 2 foods-09-01532-f002:**
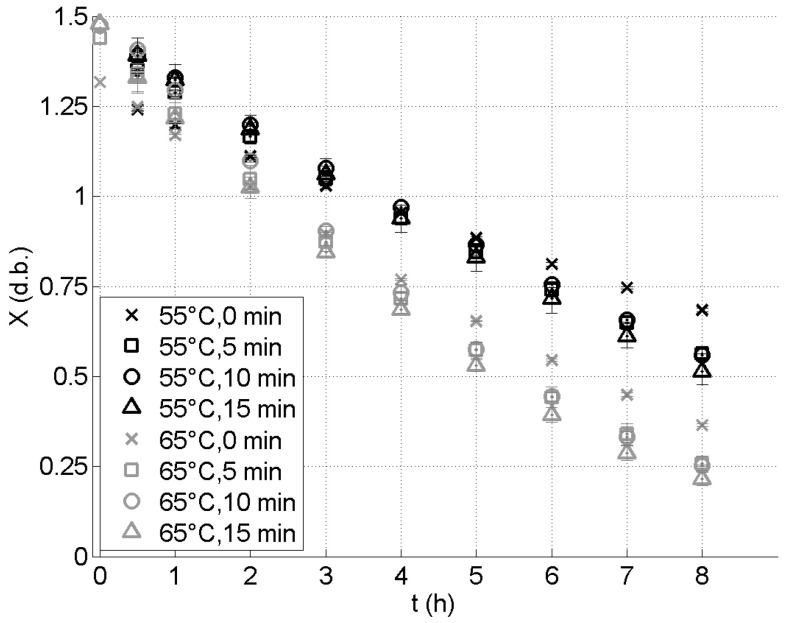
Time evolution of the moisture content of the grains during their drying, at $55{\;}^{{}^{\circ}}$C (black symbols) and $65{\;}^{{}^{\circ}}$C (gray symbols), and for various boiling times. The error bars are the standard deviations.

**Figure 3 foods-09-01532-f003:**
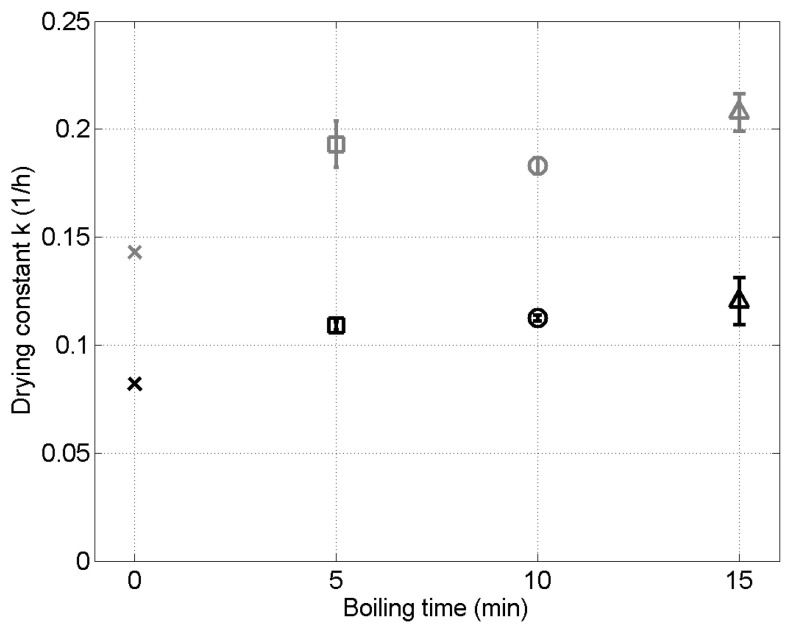
Kinetic constant of the drying *k* as a function of the boiling time (0 = no boiling), for the two drying temperatures: $55{\;}^{{}^{\circ}}$C (black symbols) and $65{\;}^{{}^{\circ}}$C (gray symbols). The error bars are the standard deviations.

**Figure 4 foods-09-01532-f004:**
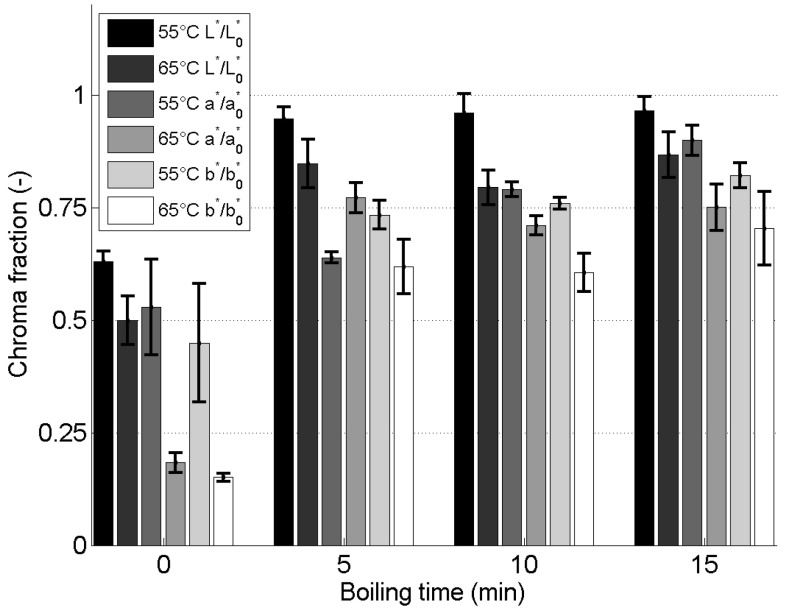
Ratios of the values of the parameters $L^{*}$, $a^{*}$ and $b^{*}$ for the grains after drying to the values of these parameters for the fresh grains (i.e., unboiled and undried), $L_{0}^{*}$, $a_{0}^{*}$ and $b_{0}^{*}$, as functions of the boiling time and for the two drying temperatures: $55{\;}^{{}^{\circ}}$C and $65{\;}^{{}^{\circ}}$C. The error bars are the standard deviations.

**Figure 5 foods-09-01532-f005:**
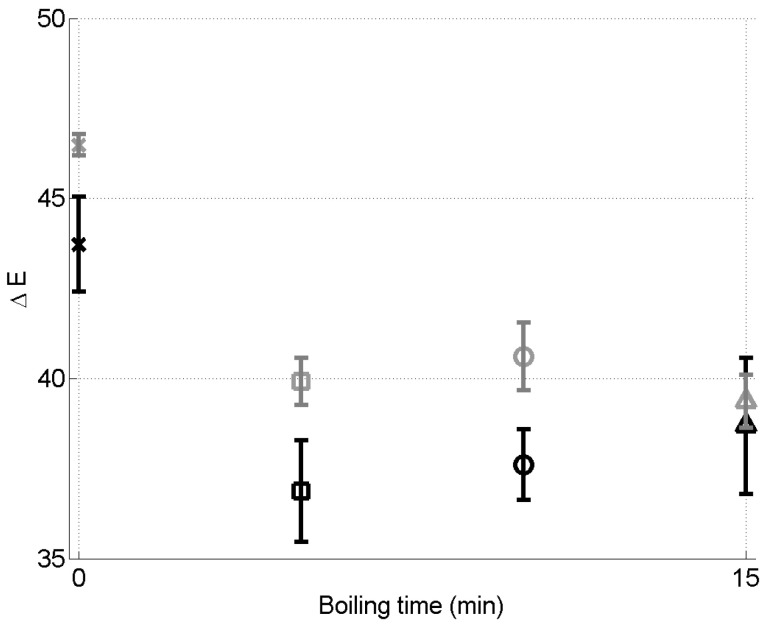
ΔE as a function of the boiling time, for the two drying temperatures: $55{\;}^{{}^{\circ}}$C (black symbols) and $65{\;}^{{}^{\circ}}$C (gray symbols). The error bars are standard deviations.

**Table 1 foods-09-01532-t001:** Properties of fresh (i.e., unboiled and undry) Kampot mature pepper. $X_{0}$, the piperine content, total phenolic content (TPC), and total flavonoid content (TFC) are expressed per gram of dry matter.

Property	Average Value	Standard Deviation	Number of Samples
$X_{0}$	1.3 g/g	0.07	4
Piperine conc.	38.5 mg/g	10.7	12
TPC	9.1 mg/g	1.1	12
TFC	42.2 mg/g	6.6	12
$L^{*}$	12.6	1.4	6
$a^{*}$	3.0	0.4	6
$b^{*}$	3.5	0.2	6

**Table 2 foods-09-01532-t002:** Ratio of the piperine content of dry grains to the piperine content of fresh grains, as a function of the boiling time, for the two drying temperatures: $55{\;}^{{}^{\circ}}$C and $65{\;}^{{}^{\circ}}$C.

	Drying at $55{\;}^{{}^{\circ}}C$		Drying at $65{\;}^{{}^{\circ}}C$	
Boiling Time	Average Value	Standard Deviation	Average Value	Standard Deviation
0 min	1.22	0.28	1.28	0.04
5 min	0.98	0.03	0.89	0.22
10 min	0.96	0.05	1.04	0.10
15 min	1.07	0.12	1.02	0.17

**Table 3 foods-09-01532-t003:** Ratio of the TPC of dry grains to the TPC of fresh grains, as a function of the boiling time, for the two drying temperatures: $55{\;}^{{}^{\circ}}$C and $65{\;}^{{}^{\circ}}$C.

	Drying at $55{\;}^{{}^{\circ}}C$		Drying at $65{\;}^{{}^{\circ}}C$	
Boiling Time	Average Value	Standard Deviation	Average Value	Standard Deviation
0 min	0.49	0.30	0.44	0.05
5 min	1.36	0.16	1.04	0.28
10 min	1.27	0.10	0.96	0.20
15 min	1.37	0.31	0.89	0.16

**Table 4 foods-09-01532-t004:** Ratio of the TFC of dry grains to the TFC of fresh grains, as a function of the boiling time, for the two drying temperatures: $55{\;}^{{}^{\circ}}$C and $65{\;}^{{}^{\circ}}$C.

	Drying at $55{\;}^{{}^{\circ}}C$		Drying at $65{\;}^{{}^{\circ}}C$	
Boiling Time	Average Value	Standard Deviation	Average Value	Standard Deviation
0 min	0.40	0.01	0.48	0.06
5 min	1.02	0.07	1.20	0.14
10 min	1.33	0.13	1.36	0.09
15 min	1.36	0.11	1.25	0.20

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
