# Peer review of "Experimental Characterization of the Drying of Kampot Red Pepper (Piper nigrum L.)"

_foods, 2020, doi:10.3390/foods9111532_

Round 1

Reviewer 1 Report

The authors improved the manuscript and explained all the doubts raised in the previous review.

Author Response

Dear Reviewer,

Many thanks for this positive review.

Best regards,

The authors

Reviewer 2 Report

The author has revised.

Author Response

(The authors gave the same response as above.)

Reviewer 3 Report

This paper lacks a good experimental design, as more drying replicates are needed regarding the high standard deviation observed in the quality parameters. 

The author' answer does not make sense "Thus, further drying experiments would not reduce the dispersion." From a statistical point of view, if one increases the number of replicates, one has more robust statistical tests. Furthermore, the review paper referred by the author does not mention "inherent variability" or high standard deviation in the Folin-Ciocalteu assay.

I also noticed that the paper has other small lapses , that are not common, particularly in a 6 authors paper. 

Is also needs improvement in grammar and lexicon.

Author Response

Dear Reviewer, 

Many thanks for your feedback on our paper. We have tried to address your comments. 

First, you pointed out that the English of the article needed to be improved. To do this, a native speaker, who is also an English teacher, proofread the article and many small grammar and spelling corrections were made to it. Therefore, we think that, now, the English of the paper meets the standard for publication in Foods.

Second, to answer your comment about the experimental methodology, we have tried to give more detail in the text (lines 303 – 329 + two additional tables in appendix) about this methodology and the results, mainly to explain where does this high dispersion come from, but also to compare it to other results and to discuss its implication on the significance of the results. Notably, we insist on the fact that the important standard deviation of the results regarding the TPC does not come from the limited number of drying tests. It rather comes from the fact that, when after a single drying experiment the TPC measurement is repeated six times on the resulting extract, the results obtained show a large dispersion. It is due to the difficulty of applying the Folin-Ciocalteu method in the lab in Cambodia (no air conditioning, large variation of the room temperature and humidity within hours...). This affirmation is supported by the two new tables in the appendix (Tables A1). They provide, for each drying test at 65°C and for each boiling time, the average and the standard deviation of the 6 evaluations of the TPC ratio. These two tables can be compared with Table 3 in the core of the text (its right part) that gives, at 65°C and for each boiling time, the average and the standard deviation of the 12 evaluations of the TPC ratio (so when the results from the two tests are analyzed together). It can be observed that these three tables are quite close to each other. It is because, when an experiment is performed, the drying conditions (boiling duration, drying temperature, ventilation rate) are very well controlled and the experiment involves approximately 300 grains (making the extract independent of the natural variability of the product). This good control of the conditions and this independence on the natural variability of the product (plus the high precision of mass and time measurements) explain why, in Figures 2 and 3, the dispersion of the results regarding the drying kinetics is very small. So, basically, we can expect that the extracts obtained at the end of the two drying experiments with the same drying conditions are really close to each other. Consequently, for given drying conditions, an additional drying experiment is not likely to reduce the dispersion. Of course, it would give 18 values of the TPC instead of 12 and make the statistical tests more robust but, despite this dispersion on the TPC results and as presented in the paper, the statistical analysis of the results of the two drying experiments allows drawing significant conclusions.

Additionally, as suggested by the editor, we have added a picture of Kampot pepper grains and some information about their size and weight. The two other reviewers had no remark on the revised version of the paper.

With this, we hope that, now, the paper meets the high quality standards for publication in Foods.

Best regards,

Benoit Haut

Round 2

Reviewer 3 Report

I have no further suggestions and accept it in present form.

This manuscript is a resubmission of an earlier submission. The following is a list of the peer review reports and author responses from that submission.

Round 1

Reviewer 1 Report

The drying procedure is unclear. What was the end parameter of the drying process (time, moisture content of the material)? What does it mean that "the sample was left in the oven until complete drying" - for how long, at what temperature, to what humidity? For what purpose, samples dried convectively were subsequently freeze-dried? This point is critical. Please clarify in detail the drying protocols. Please provide the equilibrium moisture content.

Which samples were subjected to qualitative analysis? After convection or freeze-drying?
How were X and Xf determined? I guess the same as X0 but what mass was then substituted in the denominator?

Figures 5 and 6 are illegible, suggests to present the data in the tables?

Why such large dispersion (standard deviations) is present for color, TPC and TFC? The authors mention that they conducted an analysis of variance (ANOVA) but did not provide its results. Have you done any posthoc tests?

Author Response

Best regards,

Benoit Haut 

Reviewer 2 Report

Why freeze drying was used if the material was dried to solid mass?

Complete information on color measurement methodology (observer, light source).
Was the pepper ground before measuring the color? what was the thickness of the layer?

complete information on the boilng process. what were the proportions of water to pepper

Provide information on the number of replicates of all analytical procedures

Perform a statistical analysis of the results obtained. Presenting only the standard deviation is not enough

Author Response

Best regards,

Benoit Haut 

Reviewer 3 Report

The main issue to accept this paper is that it is necessary another replicate for the drying experiments at the two different temperatures. Further comments are included in the attached file.

Author Response

Best regards,

Benoit Haut 
